# Exploring Factors Associated with Health Status and Dietary Supplement Use Among Portuguese Adults: A Cross-Sectional Online Survey

**DOI:** 10.3390/healthcare13070769

**Published:** 2025-03-30

**Authors:** Sandra Leal, Ana Catarina Sousa, Rui Valdiviesso, Inês Pádua, Virgínia M. F. Gonçalves, Cláudia Ribeiro

**Affiliations:** 1Associate Laboratory i4HB-Institute for Health and Bioeconomy, University Institute of Health Sciences-CESPU, 4585-116 Gandra, Portugalmaria.silva@iucs.cespu.pt (I.P.); virginia.goncalves@cespu.pt (V.M.F.G.); claudia.ribeiro@iucs.cespu.pt (C.R.); 2UCIBIO-Applied Molecular Biosciences Unit, Toxicologic Pathology Research Laboratory, University Institute of Health Sciences (1H-TOXRUN, IUCS-CESPU), 4585-116 Gandra, Portugal; 3Department of Sciences, University Institute of Health Sciences-CESPU, 4585-116 Gandra, Portugal; rui.santa@iucs.cespu.pt; 4RISE-Health, Faculty of Nutrition and Food Sciences, 4150-180 Porto, Portugal; 5UCIBIO-Applied Molecular Biosciences Unit, Translational Toxicology Research Laboratory, University Institute of Health Sciences (1H-TOXRUN, IUCS-CESPU), 4585-116 Gandra, Portugal; 6UNIPRO-Oral Pathology and Rehabilitation Research Unit, University Institute of Health Sciences (IUCS-CESPU), 4585-116 Gandra, Portugal

**Keywords:** multivitamin–mineral supplements, self-health perception, self-care practices, medication use, sociodemographic factors, health-related risk

## Abstract

**Background/Objectives**: Dietary supplements are associated with general well-being. However, there is a growing concern about health risks from unlabeled harmful substances, contaminants, or their interactions with conventional drugs. The use of dietary supplements should also be monitored in vulnerable groups. Hence, this study aimed to explore key factors associated with dietary supplement use and health status among Portuguese adults. **Methods**: An online-based cross-sectional survey was conducted to assess health, medication, and dietary supplement use. Data were collected from January to February 2023. The participants (N = 449) were categorized into age groups (18–29, 30–39, 40–60, and 60+). Descriptive and multivariate statistical analysis were performed. **Results**: Overall, 73% of the participants were female, with 38% in the 40–60 age group. Among male participants (27%), more than half belonged to the two older age groups. Additionally, 42% reported having a diagnosed disease, 43% reported using medication, and 66% rated their health as “good”. Participants aged 60+ with multiple diagnosed diseases and a higher BMI were associated with poorer self-rated health. Cardiovascular drug use was more prevalent among males and the 60+ age group, positively correlating with the number of medications (r = 0.40, *p* < 0.001). Medication use (OR = 0.25, 95%CI: 0.10–0.60) and diagnosed diseases (OR = 0.34, 95%CI: 0.17–0.69) were associated with a lower likelihood of avoiding supplement use. Females were more likely to use supplements, with multivitamin–mineral supplements most common among both sexes and adults aged 60+. **Conclusions**: As supplement use increases, developing evidence-based guidelines for safe use, especially among vulnerable populations, is crucial.

## 1. Introduction

Dietary supplements and herbal infusions are widely used for self-care practices to manage health and promote well-being, increasing in use over the past two decades [1,2,3]. These products include vitamins, minerals, omega-3 fatty acids, botanical extracts, and probiotics [4,5,6] and are often considered part of complementary and alternative medicine (CAM), with many believing them to offer benefits in areas such as well-being, weight management, immunity, and cognitive function [2,7,8,9].

The prevalence of dietary supplement use varies across countries, with higher rates estimated in the United States and lower but steadily increasing trends in Europe [2,4,6,10,11,12,13]. Notably, multivitamin–mineral supplements remain the most commonly used across populations in many countries, yet regional variations exist, influenced by sociodemographic and health-related factors [10,12,13,14,15].

Despite their potential benefits, dietary supplements should be used cautiously as they do not replace a balanced diet and may pose risks, particularly when self-prescribed [4,6,16,17]. Furthermore, concerns about poor quality control in production and inconsistencies in label information, such as missing or undeclared ingredients, pose public health risks [18]. Many dietary supplement users believe that these products may promote overall health or reduce risks for some medical conditions such as cardiovascular diseases (CVDs), hypercholesterolemia, depression [12,19,20], liver diseases [21], or inflammatory bowel diseases [22]. However, the effectiveness of dietary supplementation remains a topic of debate, with limited evidence supporting its beneficial effects.

Additionally, concerns are growing about their interactions with prescription medications, which may lead to reduced efficacy or harmful side effects [23,24,25]. For instance, certain supplements, such as omega-3 fatty acids and herbal infusions, may interfere with anticoagulants and antihypertensive drugs [25,26,27,28], underscoring the need for careful monitoring when used alongside prescribed medications. Furthermore, there is a rising trend in the use of supplements and herbal medicines among patients with chronic CVD or cardiovascular risk factors, often in combination with prescribed cardiovascular medications [29,30,31]. The self-prescribed nature of dietary supplements [1,15,32,33] raises safety concerns, as misuse can pose significant risks to both patients and the general population [27,28,30,32,34]. A recent study has shown that, inappropriate medication use is common among the Portuguese older population and tends to increase with the number of diagnosed diseases [35]. However, it does not address the gaps in understanding the concurrent use of dietary supplements and medications in this population. Despite the growing use of dietary supplements, particularly among older adults, key questions remain unanswered regarding the factors influencing their use and the potential risks, including interactions with prescribed medications. Addressing these gaps is critical for developing evidence-based public health recommendations to ensure safe and effective use.

Recent studies have explored the sociodemographic and health-related determinants of dietary supplement use, particularly in Europe, showing that supplement use is more prevalent among females, older adults, and individuals with health conditions [2,12,36]. In Portugal, similar trends have been observed in the use of dietary supplements by the general population, with higher prevalence of dietary supplement use among females [37]. In addition, multivitamin and mineral supplements are more frequently used by adults aged 18–64 and by elderly individuals, as reported by data from the National Food, Nutrition, and Physical Activity Survey 2015–2016 [37]. However, research on the specific factors of dietary supplement use in the Portuguese population remains limited, particularly regarding its relationship with self-care practices, health perceptions, and chronic disease management [8,37,38,39]. Understanding these relationships is crucial for evaluating the broader impacts of dietary supplements on population health. A 2023 study involving 1144 Portuguese adults found that about 27% reported using dietary supplements. Higher consumption was associated with higher education levels, greater nutritional knowledge, and healthier lifestyles, including regular physical activity [40]. Furthermore, while there is no specific information on potential interactions between medications and dietary supplements in older adults, a recent study highlighted that inappropriate medication use is a common issue among this population in Portugal [35]. However, comprehensive data on dietary supplement use across the general population remain limited, underscoring the need for further research in this area.

Therefore, the aim of this study was to explore the prevalence and patterns of dietary supplement use, alongside medication use and health perception, in a sample of adults in Portugal. Specifically, we aimed to investigate the factors influencing dietary supplement use and the potential associations with self-perceived health, medication use, and sociodemographic characteristics.

## 2. Materials and Methods

### 2.1. Study Design and Participants

This cross-sectional study used data collected from an online survey conducted between January and February 2023. This survey encompassed multiple sections that assessed various health-related behaviors. In the present study, we focused on analyzing data related to health status, self-care practices (such as medication use and dietary supplementation), and sociodemographic factors. The detailed study design and findings from this dataset, which cover other aspects of health-related behaviors, have been published in Sousa et al. [41]. In brief, the survey was designed using Google Forms^®^ and was targeted at adults aged 18 years or over. The survey included respondents from all regions of Portugal; the sample was self-selected, employing a convenience sampling method and not being representative of the general Portuguese population. Inclusion criteria required participants to be at least 18 years old; residents of Portugal; and willing to voluntarily participate. Exclusion criteria included individuals who did not complete the first part of the survey, as well as those with duplicate or inconsistent responses. Informed and extended consent was obtained at the beginning of the survey, in compliance with the Ethical Principles for Medical Research outlined in the Declaration of Helsinki and national legislation. No identifiable information was collected, and participants were explicitly informed about the study’s objectives, data confidentiality, voluntary participation, and their right to withdraw at any time. Tacit consent was assumed upon survey completion. The study received ethical approval from the Instituto Universitário de Ciências da Saúde Research Ethics Committee (reference number: CE/IUCS/CESPU-03/23).

### 2.2. Data Collection

The survey was conducted in Portuguese and included multiple sections, focusing on sociodemographic information (e.g., age, sex, education level) and health status, which encompassed self-perceived health, medication use, dietary supplementation practices, and herbal infusion consumption (consumer vs. non-consumer). Due to the sampling method, the collected data are underrepresented or overrepresented by certain demographic groups, such as participants’ education levels. The questionnaire utilized a combination of closed-ended, multiple-choice (single- and multiple-response options), and open-ended questions, adapted from previous national and international surveys [37,42,43,44]. A total of 786 participants completed the survey. During the data cleaning process, invalid responses (e.g., incomplete sociodemographic data or implausible answers) (N = 12) and responses with missing data on dietary supplementation (N = 325) were excluded from analysis, resulting in a final sample of 449 valid questionnaires (Figure 1).

### 2.3. Variables

The following sociodemographic variables were analyzed: participants’ sex (males, females), age (categorized into four groups: 18–29, 30–39, 40–60, and 60+years old), and educational level (basic, secondary, post-secondary, and higher).

Self-perceived health was assessed as part of health status characterization. Respondents rated their physical health using a 5-point scale with the following options: very good, good, fair, poor, or very poor. Based on participants’ responses, the options “poor” and “very poor” were selected by approximately 1% of participants. To ensure statistical robustness, the variable was further categorized into “very good”, “good”, and “fair” for analysis. Participants were asked if a physician had ever diagnosed them with any common medical conditions using a checklist question. The listed conditions included the following: (1) heart disease (e.g., angina pectoris, myocardial infarction, arrhythmia, heart failure); (2) stroke; (3) cancer (any type); (4) diabetes (type 1 or type 2); (5) liver disease (e.g., fatty liver, cirrhosis, hepatitis); (6) kidney disease; (7) hypertension; (8) dyslipidemia; (9) gastrointestinal disorders (e.g., Crohn’s disease, celiac disease, gastritis, ulcer); and (10) depression. Additionally, two extra response options were available: “no known illness” and “other”.

Based on participants’ responses, multiple variables were derived to characterize health status [45]. A dichotomous variable for “diagnosed disease” was created, categorizing respondents as “Yes” if they reported at least one medical condition and “No” if they reported none. Additionally, a separate binary variable for mental health conditions was created, categorizing participants as “Yes” if they selected “depression” and “No” for all others. A variable labeled “diagnosed diseases, N” was also created to reflect the number of reported medical conditions, with categories coded as “1”, “2”, and “3” (for ≥3 diseases). For participants reporting exactly one diagnosed condition (excluding depression), a categorical variable was generated to specify the type of condition, allowing for stratified analysis based on specific health conditions.

Body mass index (BMI) was calculated as weight (kg) divided by the square of height (m^2^) based on self-reported data [46,47]. The participants were categorized according to their BMI (underweight: BMI < 18.5 kgm^−2^; normal: 18.5–24.9 kgm^−2^; overweight: 25–29.9 kgm^−2^; and obese:≥30 kgm^−2^) [48].

Participants were asked if they had taken any medication prescribed by a physician within the past month (Yes/No). Medication use was assessed using a checklist question, which included the following categories: antihypertensives, antidyslipidemics, anticoagulants, contraceptives, insulin, and “other” (with a free-text field for specification). Based on participants’ responses, medication use was quantified by the total number of reported medications (coded as “1”, “2”, and “3” for ≥3) and further classified into therapeutic categories: cardiovascular, central nervous system, endocrine, digestive system, cancer-related, and other medications. Responses from participants who selected the option “other medications” and specified the medication used were reviewed to determine whether the medication fit into predefined therapeutic categories. Medications that matched an existing category were included in the appropriate group for analysis. Medications that did not correspond to any of the predefined categories were recorded as “other” and excluded from further analysis in relation to specific therapeutic categories.

Participants were queried regarding their use of dietary supplements, categorized as multivitamin–mineral supplements and/or other dietary supplements (including omega-3 fatty acids, probiotics, spirulina, or melatonin). The checklist question allowed participants to select one or both supplement categories, with additional response options for “no dietary supplement use” and “other”, which included a space for further specification. Responses were coded as follows: “no use” for no supplement consumption, “1” for multivitamin–mineral supplement use, “2” for other dietary supplement use, and “3” for the use of both supplement types. Participants who selected “other” and specified supplements such as creatine or protein powder had their responses reviewed and classified into the predefined categories when applicable. Additionally, a dichotomous variable (yes/no) was created to assess the concurrent use of dietary supplements and herbal infusion consumption.

### 2.4. Statistical Analysis

Descriptive statistics were used to summarize categorical variables as frequencies. Univariate associations between sociodemographic characteristics, health status, and self-care practices were assessed using the chi-square test (χ^2^) and Cramér’s V for the strength of the relationship. Pearson’s correlation analysis was conducted to evaluate the relationship between health perception and other health variables, adjusting for mental health conditions. Additionally, the correlation between the number of medications used and medication type was analyzed. Estimated marginal means (EMMs) were calculated to assess sex differences in dietary supplement use.

Multinomial logistic regression assessed factors associated with self-perceived health (good, very good, or fair), with independent variables including sociodemographic factors, health status (BMI, number of diagnosed diseases), and number of medications use. Another multinomial logistic regression model examined factors associated with the type of dietary supplements used (multivitamin–mineral, other supplements, both types, or none), with multivitamin–mineral supplements as the reference level. Interaction terms between variables (e.g., sex × age, medication × diagnosed disease) were included to capture potential confounding effects on supplement use. All models were assessed using the likelihood ratio test, and the results were reported as odds ratios (ORs) with 95% confidence intervals (CIs). Statistical significance was set at *p* < 0.05. Analyses were performed using Jamovi software (version 2.6.17).

## 3. Results

### 3.1. Sociodemographic Characteristics

Overall, 73% of participants were female and 27% were male. The largest age group of participants was the 18–29 age group (39%), followed by the 40–60 age group (34%). Notably, 80% had reached higher education, a proportion significantly higher than that observed in the general Portuguese population (Appendix A). Figure 2 provides an overview of the sociodemographic characteristics by sex. The highest proportion of participants was in the 18–29 age group (45% male, 37% female), while the lowest was in the 60+ age group (12% male, 6% female).

### 3.2. Health Status and Self-Care Practices Stratified by Sex

Of the 449 participants, 36% reported having a diagnosed disease, with 73% of them indicating one medical condition (Table 1). Additionally, 29% reported mental health conditions, 63% were categorized as having a normal BMI, and 69% rated their health as “good”. Excluding mental health conditions, the most commonly reported diseases were gastrointestinal disorders (16%) and heart diseases (9%) (Appendix A).

The results of sex differences in health status, medication use, and supplement use are presented in Table 1. The prevalence of diagnosed diseases showed a significant, though weak, association with sex (Cramér’s V = 0.11, *p* = 0.017). Females were more likely to report a single diagnosed condition (77% vs. 59%), while males had a higher prevalence of three or more conditions (28% vs. 9%) (Cramér’s V = 0.22, *p* = 0.018), indicating a moderate association between sex and the number of medical conditions. Among participants with a diagnosed disease, 46 (29%) reported having a mental health condition, including 29% of females and 25% of males. No significant sex differences were observed in the prevalence of mental health conditions (*p* = 0.62).

For BMI categories, 63% of participants had a normal BMI, 25% were overweight, 10% were obese, and 2% were underweight (Table 1). A significant sex difference was observed (Cramér’s V = 0.22, *p* < 0.001), with males being more likely to be classified as overweight (38%) or obese (14%) compared to females (20% and 8%). Conversely, a higher percentage of females were within the normal BMI range (70% vs. 46%).

In terms of self-perceived health, the majority of participants rated their health as “good” (69%) or “very good” (18%), with only 12% rating it as “fair.” No significant difference in self-perceived health was found between sexes (*p* = 0.91).

Regarding self-care practices, 25% of participants reported using medication, 51% used dietary supplements, and 92% consumed herbal infusions (Appendix A). A significant sex difference was observed in medication use, with more females (30%) reporting use compared to males (12%) (Cramér’s V = 0.19, *p* < 0.001), indicating a weak association between medication use and sex (Table 1).

In terms of the number of medications used, 74% of participants used one medication, 18% used two, and 9% used three or more (Table 1). The distribution of the number of medications used did not differ significantly between sexes (*p* = 0.45). However, males were more likely to report using three or more medications (14% vs. 8%), while females had a higher prevalence of using two medications (19% vs. 7%).

Dietary supplement use was more prevalent among females (59%) than males (29%) (Cramér’s V = 0.26, *p* < 0.001), indicating a moderate association between sex and dietary supplement use. Among herbal infusion consumers, no significant sex differences were observed in the proportion of individuals who also used dietary supplements (*p* = 0.23) (Appendix A). 

### 3.3. Health Status and Self-Care Practices Stratified by Age Group

Significant differences were observed in the prevalence of diagnosed diseases and the number of medical conditions across age groups (Appendix A). The overall prevalence of diagnosed diseases was lower in the 18–29 age group (27%) compared to 67% in the 60+ age group (Cramér’s V = 0.22, *p* < 0.001). The proportion of participants reporting two or more medical conditions increased with age (Figure 3a), with half of participants in the 60+ age group reporting three or more diagnosed diseases.

In contrast, 87% of participants in the 18–29 age group reported only one medical condition (Cramér’s V = 0.33, *p* < 0.001) (Appendix A). Mental health conditions showed no significant differences across age groups (*p* = 0.84).

For BMI, the 30–39 age group had the highest percentage of participants in the normal BMI category (70%), while the 60+ age group had the highest percentage of overweight participants (30%) and those classified as obese (18%) (Appendix A). However, no significant differences were observed between BMI categories across age groups (*p* = 0.14).

Self-perceived health varied significantly across age groups (Cramér’s V = 0.21, *p* < 0.001), with a moderate association observed (Appendix A). The highest proportion of individuals rating their health as “good” was found in the 30–39 age group, whereas the “fair” health category was more prevalent among those aged 60+ (Figure 3a).

Medication use varied significantly across age groups, with a weak association (Cramér’s V = 0.19, *p* < 0.001). A higher proportion of participants in the 60+ age group reported using medication (55%) compared to 22% in the 40–60 age group (Appendix A). Regarding the number of medications used (Figure 3b), the highest proportion of participants in the 18–29 age group reported using one medication, while the use of three or more medications was more frequent among participants in the 60+ age group. However, the total number of medications reported showed no significant variation across age groups (*p* = 0.33). 

Dietary supplement use also varied by age group (Figure 3b). The highest proportion of non-users was observed in the 18–29 age group, while the 60+ age group had the higher proportion of users. A significant yet weak association was found between age and dietary supplement use (Cramér’s V = 0.14, *p* = 0.028).

### 3.4. Medication and Dietary Supplement Use by Type, Sex, and Age Group

Among medications used, the most commonly used therapeutic agents were endocrine drugs (48%, primarily oral contraceptives), cardiovascular drugs (22%, antihypertensives and/or antidyslipidemics), and anticoagulants (11%) (Appendix A). Endocrine drug use was more common among female participants, with the highest usage reported in the 18–29 age group and the lowest in the 60+ age group (Appendix A). In contrast, cardiovascular and blood drugs were more prevalent among males, particularly in the 60+ age group.

The most frequently used dietary supplements were multivitamin–mineral supplements (29%), followed by the use of both supplement types (12%) and other supplements (10%), such as omega-3 fatty acids and probiotics (Appendix A). Figure 4 displays the distribution of dietary supplement types by sex (Figure 4a) and age group (Figure 4b). The prevalence of the use of dietary supplements was lower in males compared to females. Yet, male participants primarily used multivitamin–mineral supplements, with the highest prevalence observed in the 60+ age group. In contrast, females were more likely to use other supplements, especially in the 18–29 age group. Dietary supplement use showed a trend across age groups (*p* = 0.066) (Appendix A), with moderate sex differences observed (Cramér’s V = 0.26, *p* < 0.001) (Table 1).

Estimated marginal means were calculated to assess the probability of dietary supplement use across different age groups, providing a clearer understanding of age-related differences in supplement consumption (Appendix A). The results indicate that the probability of using multivitamin–mineral supplements was highest in the 60+ age group (0.45; 95%CI, 0.28–0.62) and lower in the 18–29 age group (0.30; 95%CI, 0.22–0.39). In contrast, the use of other dietary supplements was less common in the 60+ age group (0.03; 95%CI, −0.03–0.09), yet more frequent in the 18–29 age group (0.13; 95%CI, 0.06–0.20).The probability of using both supplement types remained relatively stable across age groups (0.11–0.17), with a slight increase in the 40–60 age group (0.17; 95%CI, 0.09–0.25). Non-use of supplements was more prevalent in the 18–29 age group (0.46; 95%CI, 0.37–0.54) and decreased in the 60+ age group (0.37; 95%CI, 0.20–0.54).

### 3.5. Factors Associated with Health Status and the Type of Dietary Supplements Used

The correlation analysis revealed significant associations between self-perceived health and health status factors (Appendix A). A negative correlation was observed between self-perceived health rating and the number of diagnosed diseases (r = −0.28, *p* < 0.001). Similarly, a negative correlation was found between self-perceived health and BMI categories (r = −0.20, *p* < 0.013). These relationships were observed after controlling for the presence of mental health conditions. However, no statistically significant relationship was found between self-perceived health and the number of medications used by participants (*p* = 0.53). Similarly, no significant relationships were observed between BMI and the number of medications used (*p* = 0.34).

Regarding medication use, the correlation analysis showed a significant positive correlation between the number of medications and the consumption of cardiovascular drugs (r = 0.40, *p* < 0.001) (Appendix A). In addition, a negative correlation was observed between the number of endocrine and cardiovascular drugs (r = 0.43, *p* < 0.001). These findings suggest that higher medication use is significantly associated with the use of cardiovascular drugs.

Further regression analysis clarified the significant predictors of self-perceived health (Appendix A), showing that having multiple diagnosed diseases significantly reduced self-perceived health. Participants with two diagnosed diseases were 17.26 times more likely to report their health as “fair” rather than as “good” (*p* = 0.019) and having three diseases had an even stronger effect (OR = 24.99, *p* = 0.007). Similarly, BMI had a substantial effect; overweight individuals were 17.18 times more likely to perceive their health as “fair” rather than as “good” (*p* = 0.003), and obese individuals had the strongest association with poor self-perception (OR = 61.48, *p* = 0.005). Medication use had mixed effects. Individuals using two medications were significantly less likely to perceive their health as “fair” (OR = 0.12, *p* = 0.034). However, after adjusting for confounders, those taking three or more medications showed a non-significant tendency toward poorer self-perceived health (OR = 5.84, *p* = 0.145). Neither sex (*p* = 0.710) nor age group (*p* = 0.196) significantly influenced self-perceived health. The model used showed the best fit (χ^2^ = 52.19, *p* < 0.001), indicating that it explained a substantial proportion of the variance in self-perceived health among participants.

A multinomial logistic regression analysis was conducted to assess factors associated with dietary supplement use and type among the 449 participants. The final model demonstrated a good fit (χ^2^ = 136.49, *p* < 0.001), with the likelihood ratio test indicating that sex (*p* = 0.001), medication use (*p* < 0.001), and diagnosed diseases (*p* = 0.005) were significant predictors of dietary supplement use. The unadjusted and adjusted results of the multinomial logistic regression are presented in Table 2.

Sex differences were observed, with males being more likely to avoid supplement use (OR = 2.14, *p* = 0.076), although this association did not reach statistical significance. No significant sex differences were found in the likelihood of using multivitamins or both supplement types.

Regarding age, while no clear trends emerged across other age groups (*p* = 0.281), older adults (60+ years) were 7.15 times more likely to use multivitamins compared to younger adults, although this result approached significance (*p* = 0.085).

Participants using medications were 2.35 times more likely to use multivitamin–mineral supplements over other supplements (*p* = 0.032) and were significantly less likely to avoid supplements (OR = 0.25, *p* = 0.002). Similarly, participants with diagnosed diseases were less likely to avoid supplements (OR = 0.34, *p* = 0.003).

These findings suggest that chronic diseases and obesity are key determinants of poor self-perceived health. In addition, the use of medication and diagnosed diseases were strong determinants of dietary supplement use, while age had no significant influence. Future research should explore potential subgroup differences, particularly among non-users of supplements.

## 4. Discussion

This study aimed to explore the factors influencing self-care practices, including dietary supplement and medication use, as well as the determinants of self-perceived health. Our findings reveal significant associations between dietary supplement use and sociodemographic factors, health status, and medication use among Portuguese adults, providing valuable insights into the key factors that shape their self-care behaviors.

The influence of sociodemographic factors on dietary supplement use has been well documented in several studies [2,4,10,12,29,36,37,44,49]. In line with these findings, we observed that females and older individuals (60+ age group) are more likely to use dietary supplements, while the youngest age group (18–29 years) tends to have the lowest prevalence. This pattern among Portuguese adults suggests that age plays a significant role in both dietary supplement use and non-use [14,30]. Our findings further extend the national survey data on supplementation in the Portuguese population [37], emphasizing the value of stratifying the adult population to better understand consumption behaviors [4,14].

Regarding the types of dietary supplements used, multivitamin–mineral supplements emerged as the most commonly used by both sexes and across all age groups, consistent with previous studies [2,10,13,14,37,49]. This preference may be attributed to the broad health benefits associated with multivitamins [6,29,50,51], particularly among older adults who are more likely to report chronic health conditions. However, it is important to recognize the growing trend of alternative supplements, such as omega-3 fatty acids and probiotics, which have gained popularity in recent years across the US and European countries [2,14,37]. For instance, the increasing use of omega-3 fatty acids may be driven by their association with cardiovascular benefits [26,52], yet studies fail to provide evidence supporting these optimistic effects [17,53]. Even so, these alternative supplements remain less prevalent among the Portuguese population [37], a trend corroborated by our findings. Additionally, our results reveal a sex and age pattern in the use of these products, with females and younger participants more likely to report using these dietary supplements, whereas their use was less common among those in the 60+ age group, consistent with a previous study [2,30]. These findings suggest that sociodemographic factors play a crucial role in shaping self-care behaviors, including the decision to use dietary supplements. However, the underlying reasons for these associations are complex and multifactorial, possibly reflecting cultural context or the increasing focus on preventive care in certain demographic groups [2,36,40].

The relationship between self-rated health and health status becomes even more evident as individuals age, with older adults often reporting lower self-perceived health, particularly in the presence of chronic conditions [54,55,56]. Moreover, the presence of a chronic disease is often associated with other health conditions, which may ultimately predict poorer self-rated health [54]. Regarding the influence of sex on self-perceived health, it has been suggested that differences between females and males are due to a more unfavorable health profile in females, influenced by sociodemographic factors [54,56]. For instance, in older adults, the absence of chronic disease in males predicts good health, whereas for females, the situation is more complex, suggesting that limitations in daily activities may play a significant role [57]. Despite its complexity, previous studies have shown that physical health factors such as BMI, the number of chronic diseases, and drug use can have a large effect on self-rated health, explaining much of the variation in health ratings [42,55]. In line with these findings, our results reveal a significant association between age and self-perceived health, as well as a negative correlation between self-perceived health and participants’ health status. These findings suggest a decline in perceived health in relation to aging, a higher number of medical conditions, and increased BMI. Further analysis, adjusted for potential confounders, confirmed that a higher number of diagnosed diseases, as well as overweight and obesity, had a stronger association with poorer self-rated health, while no significant effects were observed for sex or age. These findings highlight the critical role of health factors in shaping health perceptions, particularly in older adults [54,55,56].

Self-care practices, such as medication use and dietary supplement consumption, have significant implications for health status and the management of chronic conditions, including CVDs, hypertension, and gastrointestinal disorders [12,16,20,31,38,58,59]. As the global population ages, the prevalence of multiple co-existing chronic conditions among older adults is increasing [60], requiring more complex medication regimens [35,61,62]. Previous studies have shown a correlation between the number of medications, cardiovascular drugs [63], and their concurrent use with dietary supplements and herbal medicines [29,30,31,36,58,64]. Consistent with these findings, our results indicate that higher medication use among participants is significantly associated with the use of cardiovascular drugs, particularly among males and older adults. Additionally, contraceptives (the most commonly used drug among the participants of our study) have been reported in previous studies to give rise to potential risks of bleeding when used concurrently with St. John’s wort, a common ingredient in dietary supplements [36]. Our study found that, this type of endocrine drug was predominantly reported by younger age groups and females. Moreover, we observed that medication use and the presence of chronic health conditions were key determinants of dietary supplement use, with multivitamin–mineral supplements being the most commonly used type. These findings underscore the prevalent concurrent use of dietary supplements with multiple medications, particularly among older adults and those with multiple health issues, raising concerns about potential drug interactions and their impact on health [23,24,25,28,31].

Safety concerns also arise from the self-prescribed nature of practices like dietary supplements and herbal medicine, particularly among females and during pregnancy [33], with age influencing usage patterns [2,7,15,34]. For example, the consumption of herbal infusions and other herbal medicines is lower among young females but increases with age [15,41]. Additionally, the use of multiple dietary supplements increases with age, particularly among adults aged 60 and older [4,34]. Although our study did not quantify the number of supplements used, we found that herbal infusion consumers also used dietary supplements, with multivitamin–mineral and other supplements remaining stable across age groups, showing a slight increase in the 40–60 age group. This pattern of use is concerning, as dietary supplement users are often influenced by misleading health claims, which may lead to imprudent behaviors, including the misuse of supplements [2,34,49]. Additionally, many supplement users fail to disclose this information to their healthcare providers, which could have important implications for their overall health management [1,33,59,65]. These issues highlight the need for increased education on the safe use of dietary supplements, particularly among vulnerable populations, and the importance of improving communication between patients and healthcare providers to ensure optimal health outcomes.

While this study provides valuable insights, it is not without limitations. One limitation of this study is the use of convenience sampling, as participants were self-selected through an online survey. Consequently, participants aged 60+ were underrepresented, likely due to the nature of the recruitment method. Although respondents were drawn from various regions of Portugal, the lack of random sampling may have introduced selection bias, including an overrepresentation of highly educated individuals, which limits the generalizability of the findings. The high proportion of participants who attended higher education (80%) exceeds national averages and may have influenced health behaviors and dietary supplement use, which should be considered when interpreting the results. Furthermore, as this study is based on observational data collected within a specific timeframe, its design does not allow for causal inferences. Future studies employing more structured sampling methods, such as stratified or random sampling, would ensure a more representative sample. The questionnaire used to assess supplement use has not been validated; however, a previous study considered self-administered questionnaires to be a valid method for measuring dietary supplement use [66]. Information about potential confounders was collected from the questions and variables analyzed in this study, and adjustments for the included determinants were made. Nevertheless, residual confounding may still exist despite these adjustments. Another limitation is the use of self-reported weight and height to calculate BMI, which may introduce inaccuracies. To improve data accuracy, future studies should consider using objectively measured BMI values. Additionally, medications that did not fit into predefined therapeutic categories were classified as “other” and excluded from further analysis. Future research could benefit from a more comprehensive categorization of medications to capture a wider range of therapeutic agents. While herbal infusion use was considered, a detailed analysis of its interaction with dietary supplements was beyond the scope of this study. Future research should further explore this relationship and its health implications.

Nevertheless, our study is strengthened by its focus not only on the use of dietary supplements and medications as self-care practices but also on identifying the key factors influencing these behaviors and assessing the potential risks of supplement–medication interactions. These findings lay the groundwork for further research in this underexplored area. Furthermore, studies exploring subgroup differences, particularly among older adults and individuals with multiple chronic conditions or those using medications, could provide a more nuanced understanding of the factors influencing supplement use.

## 5. Conclusions and Policy Recommendations

This study highlights key factors influencing dietary supplement use among adults in Portugal and emphasizes the need for careful consideration of dietary supplement interactions with medication use. As dietary supplement use continues to rise, it is crucial to develop evidence-based guidelines to ensure safe and effective use, particularly among vulnerable populations.

Taking these findings into account, we suggest some policy–clinical recommendations:Accurate and informative labeling: Enforce the use of informative labels regarding ingredients and potential medication interactions.Adverse effects reporting: Establish a platform for reporting adverse events related to dietary supplements.Monitoring by healthcare professionals: Encourage professionals to monitor their patients’ use of dietary supplements.

Future research:Prospective epidemiological and interventional studies examining dietary supplementation in older adults and individuals with multiple health conditions are encouraged;Studies should assess healthcare professionals’ knowledge and practices regarding dietary supplements, as well as identifying gaps and barriers in healthcare–patient communication about dietary supplementation.

## Figures and Tables

**Figure 1 healthcare-13-00769-f001:**
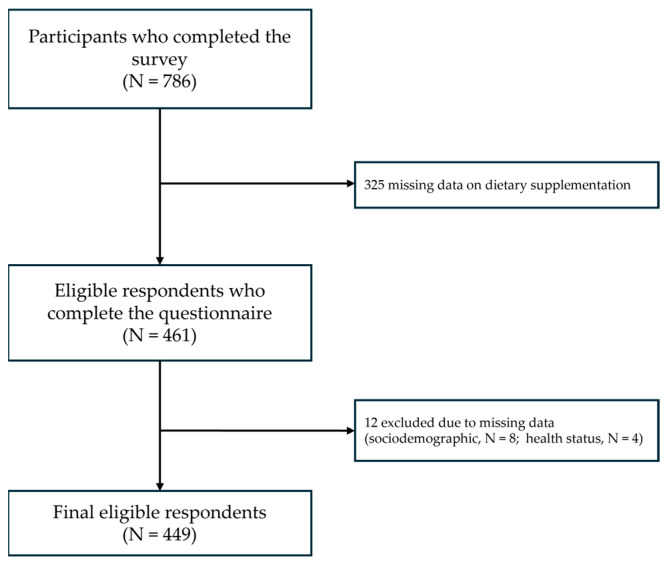
Flow chart of participants in survey.

**Figure 2 healthcare-13-00769-f002:**
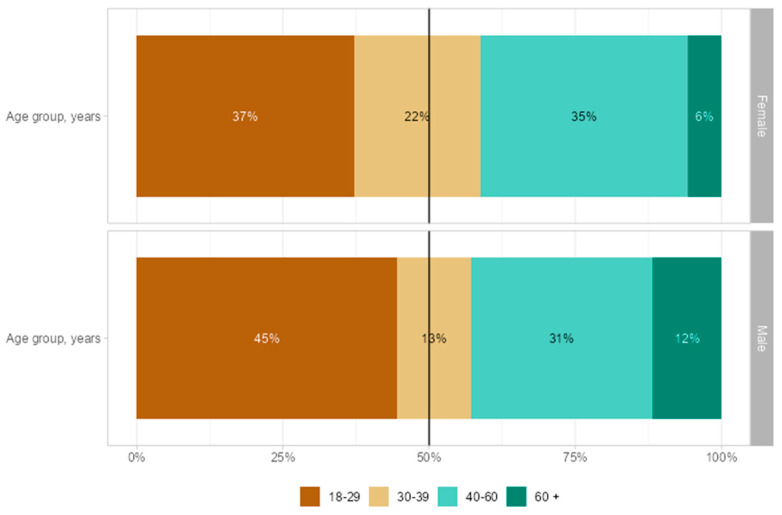
Sociodemographic characteristics of participants by sex.

**Figure 3 healthcare-13-00769-f003:**
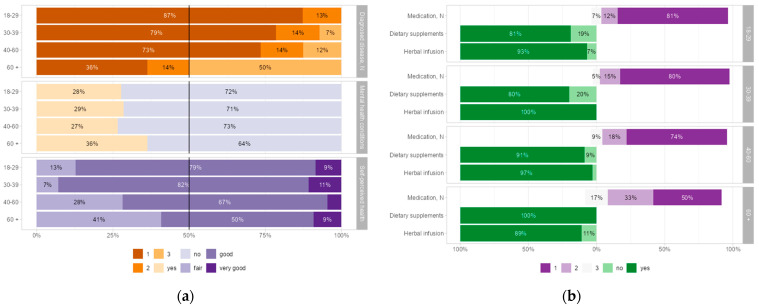
The distribution of participants’ responses across age groups using a 100% bar chart: (**a**) the number of diagnosed diseases, the presence of mental health conditions, and self-perceived health; (**b**) the number of medications used, dietary supplement use, and herbal infusion consumption.

**Figure 4 healthcare-13-00769-f004:**
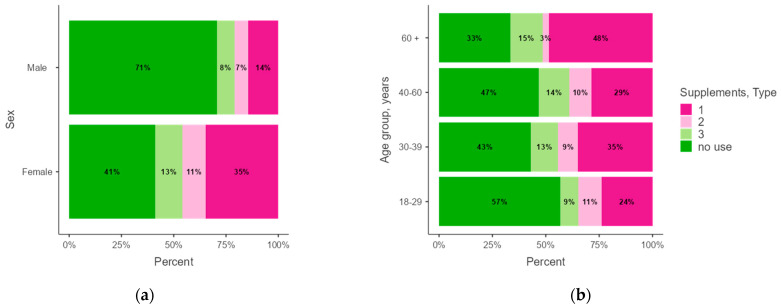
The frequency of the type of dietary supplements used by (**a**) sex and (**b**) age group. Multivitamin–mineral supplements (1); other dietary supplements (2); both supplement types (3).

**Table 1 healthcare-13-00769-t001:** Health status and self-care practices of participants, stratified by sex: Chi-square *p* values and Cramér’s V values.

	Participants				
		Sex, n (%)			
Characteristics	Total N = 449	Female 330 (73)	Male 119 (27)	*p* Value	Cramér’s V
Diagnosed disease				0.017	0.11
No	288 (64)	201 (61)	87 (73)		
Yes	161 (36)	129 (39)	32 (27)		
Diagnosed disease, N				0.018	0.22
1 disease	118 (73)	99 (77)	19 (59)		
2 diseases	22 (14)	18 (14)	4 (13		
≥3 diseases	21 (13)	12 (9)	9 (28)		
Mental health conditions				0.62	--
No	115 (71)	91 (71)	24 (75)		
Yes	46 (29)	38 (29)	8 (25)		
BMI categories				<0.001	0.22
Normal weight	284 (63)	229 (70)	55 (46)		
Overweight	110 (25)	65 (20)	45 (38)		
Obese	44 (10)	27 (8)	17 (14)		
Underweight	10 (2)	8 (2)	2 (2)		
Self-perceived health				0.91	--
Good	310 (69)	228 (69)	82 (69)		
Very good	83 (18)	62 (19)	21 (18)		
Fair	56 (12)	40 (12)	16 (13)		
Medication use				<0.001	0.19
No	335 (75)	230 (70)	105 (88)		
Yes	114 (25)	100 (30)	14 (12)		
Medication, N				0.45	--
1 medication	84 (74)	73 (73)	11 (79)		
2 medications	20 (18)	19 (19)	1 (7)		
≥3 medications	10 (9)	8 (8)	2 (14)		
Dietary supplement use				<0.001	0.26
No	220 (49)	136 (41)	84 (71)		
Yes	229 (51)	194 (59)	35 (29)		
Dietary supplement type				<0.001	0.26
(1) Multivitamin–mineral supplements	132 (29)	115 (35)	17 (14)		
(2) Other supplements	44 (10)	36 (11)	8 (7)		
(3) Both types of supplements	53 (12)	43 (13)	10 (8)		

**Table 2 healthcare-13-00769-t002:** The results of the multinomial logistic regression regarding the association of age, sex, diagnosed diseases, and medication use with the type of dietary supplement used, whereas “Other supplement types” is the reference category.

	Multivitamin–Mineral Supplement		Both Supplement Types		No Use	
	Unadjusted	Adjusted	Unadjusted	Adjusted	Unadjusted	Adjusted
	OR (95%CI)	OR (95%CI)	OR (95%CI)	OR (95%CI)	OR (95%CI)	OR (95%CI)
**Sex**						
**Male (vs. Female)**	0.67 (0.27–1.67)	0.65 (0.25–1.70)	1.05 (0.37–2.93)	1.00 (0.35–2.87)	2.78 (1.23–6.27)	2.14 (0.92–4.96)
**Age Groups, Years**						
**30–39 (vs. 18–29)**	1.70 (0.66–4.38)	1.81 (0.69–4.76)	1.74 (0.56–5.42)	1.82 (0.58–5.69)	0.88 (0.35–2.18)	0.91 (0.36–2.33)
**40–60 (vs. 18–29)**	1.24 (0.57–2.74)	1.40 (0.62–3.14)	1.74 (0.68–4.43)	1.87 (0.72–4.82)	0.85 (0.41–1.78)	0.94 (0.44–2.02)
**60+ (vs. 18–29)**	7.24 (0.89–58.63)	7.15 (0.85–60.30)	6.33 (0.67–60.17)	6.33 (0.64–62.64)	2.09 (0.25–17.16)	3.21 (0.36–28.46)
**Diagnosed Diseases**						
**Yes (vs. No)**	1.03 (0.52–2.04)	0.70 (0.34–1.47)	0.96 (0.43–2.14)	0.75 (0.32–1.76)	0.26 (0.13–0.52)	0.34 (0.17–0.69)
**Medication Use**						
**Yes (vs. No)**	2.38 (1.15–4.96)	2.35 (1.08–5.12)	1.45 (0.62–3.39)	1.50 (0.60–3.72)	0.17 (0.08–0.40)	0.25 (0.10–0.60)

## Data Availability

The data will be available upon reasonable request to the corresponding author.

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
