# Peer review of "Exploring Factors Associated with Health Status and Dietary Supplement Use Among Portuguese Adults: A Cross-Sectional Online Survey"

_healthcare, 2025, doi:10.3390/healthcare13070769_

Round 1

Reviewer 1 Report

Comments and Suggestions for Authors

I red with interest the paper titled "Self-Care Practices and Health Status: Factors Associated with Dietary Supplements Use Among Portuguese Adults".

I have minor comments that could be used to improve the manuscript. 

1. Whats the rationale of those age groups? Since they are not balanced, it creates a bias on the analysis.

2. Gender should be "sex" instead. Gender is a social construct and what was asked was sex. 

3. Line 162 needs a reference. 

4. From the analysis of Figure 1, it appear that there are a huge sample bias related with age groups, but even more evident on the education level and that create bias in the results. Around 80% of the respondents have higher education level, while in Portugal, that level 31% of active population (Eurostat, 2023). The reasons for that must be discussed (sampling technique with online survey are usually reported to be answered by the motivated/interested users vs non-motivated). 

Author Response

Reviewer 1

Comments and Suggestions for Authors - I red with interest the paper titled "Self-Care Practices and Health Status: Factors Associated with Dietary Supplements Use Among Portuguese Adults". I have minor comments that could be used to improve the manuscript.

Response: The authors are grateful to the Reviewer for the constructive comments and careful reading of this manuscript that helped us to increase the quality of the manuscript. We revised all sections of the manuscript according to the reviewers’ suggestions and comments, please kindly see, the revised version of the manuscript.

1 - Whats the rationale of those age groups? Since they are not balanced, it creates a bias on the analysis.

Response: The authors thank the reviewer for this observation and clarify. In fact, Portugal's national data on supplement use reports on the adult population aged 18–64; however, there is limited information on whether supplement use patterns vary among different adult age groups (e.g., young adults and more older individuals). Therefore, stratifying the adult Portuguese population would help clarify variations in supplement use across different age groups, which is an approach commonly adopted in other published studies evaluating the influence of age on supplement use.

For instance, Soukiasian et al. (2022) [Soukiasian et al. Nutrients 2022, 14, 5131, doi:10.3390/nu14235131] assessed a large sample of Greek individuals, although it was not representative of different age groups. This study as well as others, demonstrated that the motivations for use or not dietary supplements vary by age group. For example, Soukiasian et al. found that young men who use supplements often have fitness-related goals, whereas older adults are more likely to take supplements for health or illness-related reasons.

2 - Gender should be "sex" instead. Gender is a social construct and what was asked was sex.

Response: The authors are grateful to the Reviewer for the insightful comment. We carefully considered this suggestion. We revised all sections of the manuscript according to the reviewers’ suggestions, please kindly see the revised version of the manuscript.

3 - Line 162 needs a reference.

Response: The authors thank the Reviewer for bringing this to our attention. The reference related to BMI categories was added to Material and Methods section. Please see line 177.

4 - From the analysis of Figure 1, it appear that there are a huge sample bias related with age groups, but even more evident on the education level and that create bias in the results. Around 80% of the respondents have higher education level, while in Portugal, that level 31% of active population (Eurostat, 2023). The reasons for that must be discussed (sampling technique with online survey are usually reported to be answered by the motivated/interested users vs non-motivated).

Response: The authors thank the Reviewer for this valuable comment. We revised the manuscript to clarify this limitation of the study design. Also, Figure 1 (now Figure 2) was adjusted to remove this variable. Please see:

Lines 497—501: “the lack of random sampling may have introduced selection bias, including an overrepresentation of highly educated individuals, which limits the generalizability of the findings. The high proportion of participants with higher education (80%) exceeds national averages and may have influenced health behaviors and supplement use, which should be considered when interpreting the results.”

Reviewer 2 Report

Comments and Suggestions for Authors

Thank you for the opportunity to review this manuscript. The manuscript “Self-care practices and health status: Factors associated with dietary supplements use among Portuguese adults” presents an online cross-sectional study that investigates and studies the prevalence and patterns of dietary supplement use and health status in adults. The study is interesting, however, considering the broader context of presented studies and further impact of the published papers, I have some corrections /suggestions suggested to be addressed before next steps in the publication.

  1. In the introduction section of the manuscript, I would suggest including packaging details about the herbal products i.e., labeling, product information etc.
  2. I would also suggest expanding the introduction section further to include information about country statistics about the use of herbal products and details about the health complication for which these substances are mostly used.
  3. The authors have included individuals 18 years or above. However, it is reported that these dietary supplement use and products are more common aged ≥40 years?
  4. The authors have not provided information about the participation rate of the study. How many participants were contacted? And how many agreed to participate in the study. This information is helpful in understanding the potential selection/participation bias. Also, participation rate of both genders will be helpful.
  5. The authors have well documented limitations of the study in the discussion section of the manuscript. However, one of the major limitations associated with the study design i.e., cross-sectional survey is not described. I would suggest including this limitation.

Minor comments:

  1. In the abstract of the manuscript, the authors have mentioned “mislabeled harmful substances”. Are these herbal substances “mislabeled”? OR “unlabeled”. Mostly, these herbal substances are unlabeled.
  2. In the abstract section the authors have mentioned “positively correlating with the number of medications” I would recommend including quantitative value for the positive association to know the exact amount of correlation.
  3. I would suggest replacing “dietary supplements use”, “Health status” and “Self care practices” in key words with suitable similar words. As these words have already been used in the title of the manuscript.

Author Response

Comments and Suggestions for Authors - Thank you for the opportunity to review this manuscript. The manuscript “Self-care practices and health status: Factors associated with dietary supplements use among Portuguese adults” presents an online cross-sectional study that investigates and studies the prevalence and patterns of dietary supplement use and health status in adults. The study is interesting, however, considering the broader context of presented studies and further impact of the published papers, I have some corrections /suggestions suggested to be addressed before next steps in the publication.

Response: The authors are grateful to the Reviewer for the insightful comments. We revised all sections of the manuscript according to the reviewers’ suggestions, please kindly see the revised version of the manuscript.

In the introduction section of the manuscript, I would suggest including packaging details about the herbal products i.e., labeling, product information etc. I would also suggest expanding the introduction section further to include information about country statistics about the use of herbal products and details about the health complication for which these substances are mostly used.

Response: The authors thank the Reviewer for the thoughtful comments. In response to this constructive feedback, we have revised the Introduction section. Please see:

Lines 56–58 “Furthermore, concerns about poor quality control in production and inconsistencies in label information, such as missing or undeclared ingredients, pose public health risks [18].”

Regarding more detailed information on the national trends in herbal infusion consumption, we acknowledge that this is an interesting topic. However, it extends beyond the primary objective of the present study.

The authors have included individuals 18 years or above. However, it is reported that these dietary supplement use and products are more common aged ≥40 years?

Response: The authors thank the Reviewer comment and clarify Portugal's national data on supplement use reports on the adult population aged 18–64, but there is a lack of information on whether there is an increasing trend among individuals aged 40 and older. Therefore, a stratification of the adult Portuguese population would help clarify variations in dietary supplement use across different age groups, which is an approach commonly adopted in other studies that evaluate the influence of age on supplement use.

For instance, Soukiasian et al. [Soukiasian et al. Nutrients 2022, 14, 5131, doi:10.3390/nu14235131] investigated the motivations for using supplements by age group. That study found that young men who use supplements often have fitness-related goals, whereas older adults are more likely to take supplements for health or illness-related reasons.

The authors have not provided information about the participation rate of the study. How many participants were contacted? And how many agreed to participate in the study. This information is helpful in understanding the potential selection/participation bias. Also, participation rate of both genders will be helpful.

Response: The authors thank the Reviewer for the thoughtful review and appreciate the suggestion to enhance the clarity of the Materials and Methods section. To address this concern, we have revised the section and included a new Figure 1, which presents the participant flow diagram and reinforces the description in the text. Please see page 4 of the revised manuscript.

The authors have well documented limitations of the study in the discussion section of the manuscript. However, one of the major limitations associated with the study design i.e., cross-sectional survey is not described. I would suggest including this limitation.

Response: The authors thank the Reviewer for this suggestion. We revised the manuscript to clarify the limitations of the study design. Please see:

Lines 497-501 “the lack of random sampling may have introduced selection bias, including an overrepresentation of highly educated individuals, which limits the generalizability of the findings. The high proportion of participants with higher education (80%) exceeds national averages and may have influenced health behaviors and supplement use, which should be considered when interpreting the results.”

Minor comments:

In the abstract of the manuscript, the authors have mentioned “mislabeled harmful substances”. Are these herbal substances “mislabeled”? OR “unlabeled”. Mostly, these herbal substances are unlabeled.

Response: The authors thank the Reviewer for bringing this to our attention. We apologize for it and the alteration has been made in the abstract.

In the abstract section the authors have mentioned “positively correlating with the number of medications” I would recommend including quantitative value for the positive association to know the exact amount of correlation.

Response: The authors are grateful to the Reviewer for the insightful comment. In this revised version, we included the correlation value in the abstract. Please see line 33.

I would suggest replacing “dietary supplements use”, “Health status” and “Self care practices” in key words with suitable similar words. As these words have already been used in the title of the manuscript.

Response: The authors thank the Reviewer for bringing this to our attention. We have made the necessary revisions to the keywords, aligning them with the proposed new title.

Reviewer 3 Report

Comments and Suggestions for Authors

. Title and Abstract

  • The manuscript’s title succinctly conveys the main topic—self-care practices (including dietary supplements) and health status among Portuguese adults. However, the phrase “Factors Associated” is quite broad. Consider adding brief specificity (e.g., “A Cross-Sectional Online Survey” or “An Online Study in Portugal”) to set clear reader expectations.
  • Presently, the abstract references “dietary supplements and herbal infusions” but emphasizes supplements more heavily in the results. If herbal infusion data are complementary rather than primary, clarify how they inform the overarching objective. Also, consider providing a sharper emphasis on the main statistical associations (e.g., highlight the odds ratio for medication/disease with supplement use in the abstract).
  1. Introduction
  • The introduction successfully sets a rationale for examining dietary supplement use in the Portuguese population. However, certain paragraphs reiterate global usage patterns without digging into the Portuguese context until later. Streamlining these paragraphs would help: start with global prevalence/trends, then pivot quickly to the Portuguese scenario, citing current national data or prior local studies more explicitly.
  • The mention of potential drug-supplement interactions and older adults at risk is important. However, it might help to reference whether other European or specifically Portuguese data have previously signaled these interactions as a documented concern. A sharper statement about how your study’s approach extends or differs from earlier Portuguese work would clarify the novelty.
  • The stated objective “to explore the prevalence and patterns… along with medication use and health status” is coherent. Yet, there is also mention of “self-care practices” that includes herbal infusions, but subsequent sections highlight supplement use more than infusion consumption. Ensure that the Introduction clarifies whether herbal infusions are a secondary focus or a parallel one.
  1. Materials and Methods
    • The cross-sectional, convenience sampling method is reported, but should have a more explicit limitation statement. Currently, the authors do mention selection bias. Reiterate that the sample is not nationally representative and is more skewed toward certain demographics (e.g., higher-educated participants).
    • The timeline (January–February 2023) is short; possibly note any pandemic aftermath influences if relevant.
    • Or instruments, the self-administered online questionnaire is described. There is mention that “the final sample is 449 participants” after excluding incomplete data. This is fine, yet consider a clearer flow diagram of participant inclusion/exclusion in the main text, not just in supplementary material. It would improve transparency.
    • The unvalidated nature of the supplement use questionnaire is acknowledged indirectly. Being more explicit about how the question sets were formulated (e.g., adapted from prior national/international surveys) would reassure the reader about content validity.
    • Regarding variable and coding, the process for deriving composite variables (e.g., “medication, N,” “diagnosed diseases, N,” or mental health conditions) is described in detail. This is helpful. However, the rationale for grouping certain outcomes (for instance, combining “poor” or “very poor” self-perceived health) should appear earlier. Some explanations only appear in the Results or supplementary.
    • Defining “dietary supplements” is crucial, especially since some participants selected “other.” Confirm that sports nutrition products (e.g., protein powders) or specialized nutraceuticals are included/excluded. Ensuring consistent categorization is key.
    • For the statistical approach, the choice of multinomial logistic regression for examining the type of supplement is suitable. Yet, the authors also used partial correlation analyses; this step should be justified: why partial correlations were used instead of controlling within the regression model? Clarifying rationale would avoid confusion.
    • The authors note controlling for mental health conditions in some correlations. It might be helpful to mention how many participants had depression or other mental health conditions in the main text (rather than only in tables).
  1. Results
    • The breakdown by age, gender, and education is straightforward, but the higher education rate (80%) is noted to be well above national averages. A short statement in Results (or Discussion) acknowledging this heavy skew is recommended, beyond referencing the convenience sample in Methods, to contextualize the findings.
    • Regarding the discussion of how “males used more cardiovascular medication, females used more contraceptives” , the text jumps between the medication type and the number of medications used. Consider reorganizing the medication results under subheadings for clarity: (i) prevalence of medication use, (ii) distribution by medication category, (iii) correlation with supplement use.
    • On the association with self perceived health, the results effectively highlight significant negative associations between multiple diseases/BMI and self-perceived health. However, the p-values for medication use (≥3 meds) did not appear strongly conclusive. Clarify if that is underpowered or if, after adjusting for confounders, it truly is non-significant.
  1. Discussion
    • The discussion effectively situates the high prevalence of supplement use among older adults and the potential interactions with multiple medications. Still, the manuscript can sharpen the link between your results and broader public health implications in Portugal specifically. For instance, if older Portuguese adults with multiple chronic diseases are turning more to supplements, highlight potential policy or clinical practice changes needed (counseling, systematic screening).
    • In your comparison with other literature, the discussion lumps many “international contexts.” Consider drawing more direct comparisons with neighboring southern European or other Portuguese data. The mention of “only a few previous Portuguese CAM studies” is good, but then the discussion remains somewhat general and US-focused.
    • On limitations, the sample’s overrepresentation of higher-educated individuals is acknowledged, but reemphasizing that in the Discussion would help. Another limitation not fully articulated: the cross-sectional nature restricts interpretation of whether supplement use might be a result (rather than a cause) of self-perceived poor health. A line clarifying the directionality concern would be great.
    • On the implications of your findings, the conclusion that “older adults and medication users are more likely to use supplements” is consistent with existing data but might expand on how clinicians can respond (screening for potential drug-supplement interactions, improved labelling, etc.).
    • If herbal infusion data are less central, consider acknowledging that more robust analysis on infusion-supplement interplay is beyond the scope, or integrate them more fully in the discussion if they are indeed relevant.
  1. Conclusions
  • Reiterating the core quantitative take-home (e.g., “Over half of participants used supplements; older adults used them more frequently…”) might anchor the conclusion in actual numbers.
  • The statement, “As supplement use increases, more evidence-based guidelines are crucial,” is apt but could be more robust if authors reference the need for specific research (e.g., prospective trials or intervention studies with older or multi-disease populations).
Comments on the Quality of English Language

The English could be improved to more clearly express the research.

Author Response

Reviewer 3

Response: The authors are grateful to the Reviewer for the insightful and valuable comments and suggestions. We carefully revised all sections of the manuscript according to the reviewers’ comments and suggestions, please kindly see the revised version of the manuscript.

Title and Abstract

The manuscript’s title succinctly conveys the main topic—self-care practices (including dietary supplements) and health status among Portuguese adults. However, the phrase “Factors Associated” is quite broad. Consider adding brief specificity (e.g., “A Cross-Sectional Online Survey” or “An Online Study in Portugal”) to set clear reader expectations.

Response: The authors thank the Reviewer for this valuable comment. We followed the reviewer's suggestion, and we propose an adjust to the title: “Exploring Factors Associated with Health Status and Dietary Supplements Use Among Portuguese Adults: A Cross-Sectional Online Survey”.' We are optimistic that the reviewer will agree with our proposed change.

Presently, the abstract references “dietary supplements and herbal infusions” but emphasizes supplements more heavily in the results. If herbal infusion data are complementary rather than primary, clarify how they inform the overarching objective.

Response: The authors thank the Reviewer for bringing this to our attention. Indeed, the statement was overarching, as the herbal infusion data was complementary. We apologize for this oversight. The necessary revision has been made and “herbal infusion consumption” was removed from the abstract. Please see line 22.

Also, consider providing a sharper emphasis on the main statistical associations (e.g., highlight the odds ratio for medication/disease with supplement use in the abstract).

Response: The authors are grateful to the Reviewer for the insightful comment. In this revised version, we included the odds radio and 95% CI for the association of medication and disease with supplements use. Please see lines 33-34: “Medication use (OR=0.25, 95%CI: 0.10-0.60) and diagnosed diseases (OR=0.34, 95%CI: 0.17-0.69) were associated with a lower likelihood of avoiding supplement use.”

Introduction

The introduction successfully sets a rationale for examining dietary supplement use in the Portuguese population. However, certain paragraphs reiterate global usage patterns without digging into the Portuguese context until later. Streamlining these paragraphs would help: start with global prevalence/trends, then pivot quickly to the Portuguese scenario, citing current national data or prior local studies more explicitly.

Response: The authors are grateful to the Reviewer for the insightful comment. To address this constructive comment, we have revised the Introduction section, please see pages 2 and 3. We believe that this revised version of the manuscript has enhanced its quality based on the reviewer's pertinent suggestions.

The mention of potential drug-supplement interactions and older adults at risk is important. However, it might help to reference whether other European or specifically Portuguese data have previously signaled these interactions as a documented concern. A sharper statement about how your study’s approach extends or differs from earlier Portuguese work would clarify the novelty.

Response: The authors thank the Reviewer for this valuable comment. Following the reviewer's suggestion, we have revised the entire Introduction section, incorporating new information to better clarify the scope of our study. Specifically, we have added relevant literature on dietary supplements, as well as a study addressing concerns about medication use among the Portuguese population (Campos et al., 2025; doi:10.3390/foods14050884 and Simão et al., 2019; doi:10.2147/PPA.S219346). We believe these revisions have significantly improved the manuscript's quality based on the reviewer's insightful suggestions. For details, please see pages 2 and 3, lines 72-75; 78-79; 87-99 of the Introduction.

The stated objective “to explore the prevalence and patterns… along with medication use and health status” is coherent. Yet, there is also mention of “self-care practices” that includes herbal infusions, but subsequent sections highlight supplement use more than infusion consumption. Ensure that the Introduction clarifies whether herbal infusions are a secondary focus or a parallel one.

Response: The authors thank the Reviewer for bringing this to our attention. As mentioned earlier, data on herbal infusion was not the main focus of our study. To avoid misunderstandings and to address the reviewer's concerns, we have revised the Abstract and Introduction section to provide further clarification. Please see line 22 and pages 2 and 3 of the Introduction section of the revised manuscript.

Materials and Methods

The cross-sectional, convenience sampling method is reported, but should have a more explicit limitation statement. Currently, the authors do mention selection bias. Reiterate that the sample is not nationally representative and is more skewed toward certain demographics (e.g., higher-educated participants).

Response: The authors thank the Reviewer for the valuable comments. In response to these constructive suggestions, we have revised the manuscript to clarify the limitations of the study design. Please see:

Lines 116-117 “, reflecting a convenience sampling method and not being representative of the general Portuguese population”

Lines 132-134 “Due to the sampling method, the collected data is underrepresented or overrepresented by certain demographic groups, such as participants’ education levels.”

The timeline (January–February 2023) is short; possibly note any pandemic aftermath influences if relevant.

Response: The authors thank the Reviewer for this observation. First, we would like to clarify that our observational study aimed to expand national data on dietary supplement use by capturing a specific timeframe to explore the influence of age, sex, and health status on dietary supplement consumption among adults. While the impact of COVID-19 on dietary supplement use is an important factor influencing consumption trends, it falls beyond the primary focus of our study.

Additionally, studies examining supplement use during the COVID-19 pandemic have reported significant variations across countries and time periods, making direct comparisons challenging. Also, Louca et al. (doi: 10.1136/bmjnph-2021-000250), for example, found that the prevalence of dietary supplement use during the COVID-19 pandemic remained consistent with previous national surveys.

Or instruments, the self-administered online questionnaire is described. There is mention that “the final sample is 449 participants” after excluding incomplete data. This is fine, yet consider a clearer flow diagram of participant inclusion/exclusion in the main text, not just in supplementary material. It would improve transparency.

Response: The authors are grateful to the Reviewer for the insightful comment. We carefully considered this suggestion and a new figure 1 was added in the material and methods section showing the flow diagram of participants. Please see page 4 of the revised manuscript.

The unvalidated nature of the supplement use questionnaire is acknowledged indirectly. Being more explicit about how the question sets were formulated (e.g., adapted from prior national/international surveys) would reassure the reader about content validity.

Response: The authors thank the Reviewer the valuable comment. In response to this constructive feedback, we have revised the Materials and Methods section to provide a more detailed explanation of how the questions were developed. Please see:

Lines 134-137 “The questionnaire utilized a combination of closed-ended, multiple-choice (single and multiple-response options), and open-ended questions, adapted from previous national and international surveys [37,42–44].”

Regarding variable and coding, the process for deriving composite variables (e.g., “medication, N,” “diagnosed diseases, N,” or mental health conditions) is described in detail. This is helpful. However, the rationale for grouping certain outcomes (for instance, combining “poor” or “very poor” self-perceived health) should appear earlier. Some explanations only appear in the Results or supplementary.

Response: The authors thank the Reviewer for this comment. We would like to clarify that the details requested are already provided in the manuscript. Specifically, this information can be found in Material and Methods section, on lines 139-141 [“Given that approximately 1% of participants selected “poor” or “very poor”, these response options were combined into a single category (‘fair’) for analysis to ensure statistical robustness.]. However, to ensure greater clarity, we have adjusted the sentence, please see:

Lines 153-156 “Based on participants’ responses, the options “poor” and “very poor” were selected by approximately 1% of participants. To ensure statistical robustness, the variable was further categorized into “very good”, “good” and “fair” for analysis.”

Defining “dietary supplements” is crucial, especially since some participants selected “other.” Confirm that sports nutrition products (e.g., protein powders) or specialized nutraceuticals are included/excluded. Ensuring consistent categorization is key.

Response: The authors appreciate to the Reviewer insightful comment. We have carefully considered the suggestion and included all the examples provided for participants who classified them as other types of dietary supplements (namely spirulina and melatonin). Additionally, we have included detailed information about how the option “other” was processed. In fact, some participants indicated the use of creatine and protein powers. Please see:

Lines 198-200 “Participants who selected “other” and specified supplements such as creatine or protein powder had their responses reviewed and classified into the predefined categories when applicable”.

For the statistical approach, the choice of multinomial logistic regression for examining the type of supplement is suitable. Yet, the authors also used partial correlation analyses; this step should be justified: why partial correlations were used instead of controlling within the regression model? Clarifying rationale would avoid confusion.

Response: The authors thank the Reviewer for bringing this to our attention. We apologize for the oversight in the analysis description and appreciate the suggestion which helped us improve the clarity of the Materials and Methods section.

To clarify, we performed partial correlation to examine the relationships for health perception and for medication use, without controlling supplement type. Health perception was analyzed with adjustments for mental health, considered a confounding factor, and further examined using multinomial logistic regression. The results are presented in Tables S4 and S5 of the supplementary material.

Since only 25% of participants reported medication use, distributed across different therapeutic categories, we applied this approach to investigate correlations between the number of medications used and the most frequently reported therapeutic categories. For dietary supplement use, we conducted multinomial logistic regression incorporating interactions between variables in our prediction model. To address the reviewer's concerns, we have revised the Material and Methods section to provide further clarification. Please see lines 207-211 of the revised manuscript.

The authors note controlling for mental health conditions in some correlations. It might be helpful to mention how many participants had depression or other mental health conditions in the main text (rather than only in tables).

Response: The authors thank the Reviewer the valuable comment. We have carefully considered this suggestion and revised the Results section to include information on the distribution of participants reporting on mental health conditions. Please see lines 247-249 of the revised manuscript.

Results

The breakdown by age, gender, and education is straightforward, but the higher education rate (80%) is noted to be well above national averages. A short statement in Results (or Discussion) acknowledging this heavy skew is recommended, beyond referencing the convenience sample in Methods, to contextualize the findings.

Response: The authors are grateful to the Reviewer insightful comment. To address this, we have included a statement in the Discussion section (lines 497-501) acknowledging the overrepresentation of highly educated participants and its potential impact on the study findings. This addition provides further context for interpreting the results.

Regarding the discussion of how “males used more cardiovascular medication, females used more contraceptives” , the text jumps between the medication type and the number of medications used. Consider reorganizing the medication results under subheadings for clarity: (i) prevalence of medication use, (ii) distribution by medication category, (iii) correlation with supplement use.

Response: The authors sincerely thank the Reviewer for this suggestion. However, regarding medication use, the majority of participants reported being non-users, and we did not restrict the types of medications reported, as our primary aim was to assess the overall pattern of dietary supplement use and its potential association with medication use. Therefore, the collected data on medication use and the target population do not provide detailed information on specific medication types, making it challenging to reorganize the results as suggested. Nevertheless, we revised the Results section and have made some adjustments, please see lines 316-318.

On the association with self perceived health, the results effectively highlight significant negative associations between multiple diseases/BMI and self-perceived health. However, the p-values for medication use (≥3 meds) did not appear strongly conclusive. Clarify if that is underpowered or if, after adjusting for confounders, it truly is non-significant.

Response: The authors thank the Reviewer the valuable comment. In response to this constructive feedback, we have revised the Results section to provide a more detailed explanation. Indeed, before adjusting for confounders, the use of 3 or more medications showed a statistically significant p-value. For details, please see lines 372-374 of the revised manuscript.

Discussion

The discussion effectively situates the high prevalence of supplement use among older adults and the potential interactions with multiple medications. Still, the manuscript can sharpen the link between your results and broader public health implications in Portugal specifically. For instance, if older Portuguese adults with multiple chronic diseases are turning more to supplements, highlight potential policy or clinical practice changes needed (counseling, systematic screening).

Response: The authors thank the Reviewer for this valuable comment. To clarify the following was added to the introduction:

Lines 72-79 “A recent study has shown that inappropriate medication use is common among the Portuguese older population and tends to increase with the number of diagnosed diseases [35]. However, it does not address the gaps in understanding the concurrent use of dietary supplements and medications in this population. …... Addressing these gaps is critical for developing evidence-based public health recommendations to ensure safe and effective use”

Further, in the Conclusion section a new topic was added “Conclusions and Policy Recommendations” and we suggest some policy-clinical recommendations”. Please see lines 532 – 545:

“Taking these findings into account we suggest some policy-clinical recommendations:

-             Supplement - medication interaction warnings: informative labels on dietary supplements regarding potential interaction with medication;

-             Quality and clear label information: ensure supplement quality and safety and label providing detailed information about all ingredients;

-             Promote education for healthcare professionals (clinicians, pharmacist, nutritionists, nurses among others) on supplement benefits as well as supplement – medication interactions;

-             Reporting of the adverse dietary supplement effects – platform for dietary supplement manufactures, healthcare professionals, and consumers reporting adverse events;

-             Healthcare professional monitoring: healthcare professionals following dietary supplement use of their patients.”

In your comparison with other literature, the discussion lumps many “international contexts.” Consider drawing more direct comparisons with neighboring southern European or other Portuguese data. The mention of “only a few previous Portuguese CAM studies” is good, but then the discussion remains somewhat general and US-focused.

Response: The authors thank the Reviewer the valuable comment. We have carefully considered this suggestion and revised the Discussion section to incorporate findings from previous European studies, providing greater context for dietary supplement use in this region. Please see the revised manuscript.

On limitations, the sample’s overrepresentation of higher-educated individuals is acknowledged, but reemphasizing that in the Discussion would help. Another limitation not fully articulated: the cross-sectional nature restricts interpretation of whether supplement use might be a result (rather than a cause) of self-perceived poor health. A line clarifying the directionality concern would be great.

Response: The authors thank the Reviewer insightful comments. In response to these constructive suggestions, we have revised Discussion section of the manuscript to clarify the limitations of the study design. We believe that these revisions have significantly improved the manuscript's quality based on the Reviewer's valuable feedback. Please see:

Lines 497-503 “the lack of random sampling may have introduced selection bias, including an overrepresentation of highly educated individuals, which limits the generalizability of the findings. The high proportion of participants with higher education (80%) exceeds national averages and may have influenced health behaviors and supplement use, which should be considered when interpreting the results. Furthermore, as this study is based on observational data collected within a specific timeframe, its design does not allow for causal inferences.”

On the implications of your findings, the conclusion that “older adults and medication users are more likely to use supplements” is consistent with existing data but might expand on how clinicians can respond (screening for potential drug-supplement interactions, improved labelling, etc.).

Response: The authors than the Reviewer comment and agree. Therefore, in the conclusion a new topic was added “Conclusions and Policy Recommendations” and we suggest some policy-clinical recommendations. Please see lines 532-545.

If herbal infusion data are less central, consider acknowledging that more robust analysis on infusion-supplement interplay is beyond the scope, or integrate them more fully in the discussion if they are indeed relevant.

Response: The authors thank the Reviewer insightful comment. In response, we acknowledge that a more in-depth analysis of the interplay between herbal infusion consumption and dietary supplement use is beyond the scope of this study. To clarify this, we have included a statement in the Discussion section (lines 515-518) acknowledging this limitation while maintaining the relevance of herbal infusion data in the context of self-care practices.

Conclusions

Reiterating the core quantitative take-home (e.g., “Over half of participants used supplements; older adults used them more frequently…”) might anchor the conclusion in actual numbers.

Response: The authors appreciate the Reviewer's suggestion to emphasize a quantitative take-home message in the conclusion. However, given that our study is not representative of the general population, we believe it would not be appropriate to anchor the conclusion in specific numerical findings. Instead, our study provides broader insights into supplement use patterns rather than a strictly quantitative assessment. We have ensured that our conclusion reflects this perspective while still summarizing the key findings in a meaningful way.

The statement, “As supplement use increases, more evidence-based guidelines are crucial,” is apt but could be more robust if authors reference the need for specific research (e.g., prospective trials or intervention studies with older or multi-disease populations).

Response: The authors appreciate the Reviewer's suggestion to strengthen our statement on the need for evidence-based guidelines. We have revised the manuscript to specify the importance of further research. Please see lines 547-552.

“Future research:

-             Conduct prospective epidemiological and intervention studies on dietary supplementation in older adults and individuals with multiple health conditions.

-             Investigate healthcare professionals' knowledge and practices regarding dietary supplements, as well as identify gaps and barriers in healthcare-patient communication on supplementation.”

Round 2

Reviewer 2 Report

Comments and Suggestions for Authors

Thank you for the opportunity to review the revised version of the manuscript entitled “Exploring factors associated with health status and dietary supplements use among Portuguese adults: A cross-sectional online survey”. Based on comments/suggestions and their implementation, the manuscript has improved and I have no further comments. I wish best of luck with further steps in publication.

Author Response

Your thoughtful comments have contributed to improving the clarity and quality of our study. Thank you for your support and consideration.

Reviewer 3 Report

Comments and Suggestions for Authors

The authors have done well in revising some of the issues raised in the earlier version and just some minor comments

Results

  • While you mention in the Discussion that 80% of participants have higher education, you might add a brief note in the Results section (where you present Table S1 or the relevant text) to reemphasize that this proportion is significantly high compared to the general Portuguese population. This helps readers who might skim results only.

Discussion

  • Non-representative Sampling: You have stated the convenience sampling in the Methods and later in Limitations. If possible, add one sentence clarifying if the older adult representation is proportionate (e.g., “The proportion of participants aged 60+ was relatively small, reflecting the online recruitment method...”). That helps interpret your age-related results.
  • Conclusions vs. Policy: Your “Conclusions and Policy Recommendations” section is valuable, but ensure it does not overshadow the final summary of your study. One approach is to keep the bullet-point recommendations brief and pointed.

Comments on the Quality of English Language
  • Language Polishing: The English has definitely improved. A final proofreading pass especially focusing on run-on sentences or repeated words would help. For instance, ensure that phrases like “presence of multiple diseases” or “presence of multiple diagnosed diseases” are consistently used.

  • You have mostly used “males/females,” but in a few places, “gender” creeps in. Decide on one term for consistency  for example, “sex” if referring to biological classification.

Author Response

Results

1- While you mention in the Discussion that 80% of participants have higher education, you might add a brief note in the Results section (where you present Table S1 or the relevant text) to reemphasize that this proportion is significantly high compared to the general Portuguese population. This helps readers who might skim results only.

Response: The authors thank the Reviewer insightful suggestion. To improve clarity and ensure that readers focusing on the Results section recognize this key point, we have added a brief note emphasizing that the proportion of participants with higher education is substantially higher than in the general Portuguese population. Please see lines 226-228: “Notably, 80% had attained higher education, a proportion significantly higher than that observed in the general Portuguese population”

Discussion

2- Non-representative Sampling: You have stated the convenience sampling in the Methods and later in Limitations. If possible, add one sentence clarifying if the older adult representation is proportionate (e.g., “The proportion of participants aged 60+ was relatively small, reflecting the online recruitment method...”). That helps interpret your age-related results.

Response: The authors appreciate the Reviewer’s suggestion. To address this, we have added a clarifying sentence in the Limitations section, explicitly stating that the proportion of participants aged 60+ was relatively small, likely due to the online recruitment method. This addition helps contextualize the age-related findings. Please see lines 497–498 in the revised manuscript.

“Consequently, participants aged 60+ were underrepresented, likely due to the nature of the recruitment method.”

3-  Conclusions vs. Policy: Your “Conclusions and Policy Recommendations” section is valuable, but ensure it does not overshadow the final summary of your study. One approach is to keep the bullet-point recommendations brief and pointed.

Response: The authors thank the Reviewer for this valuable comment. To maintain balance, we have streamlined the bullet-point recommendations for clarity and conciseness while ensuring they do not overshadow the study’s conclusions. We appreciate your insightful feedback. Please see lines 536-541:

“Taking these findings into account we suggest some policy-clinical recommendations:

  • Accurate and Informative Labeling: Informative labels regarding ingredients and potential medication interactions.
  • Adverse effects reporting: Establish a platform for reporting adverse events related to dietary supplements.
  • Monitoring by healthcare professionals: Encourage professionals to monitor their patients' use of dietary supplements.”

4- Language Polishing: The English has definitely improved. A final proofreading pass especially focusing on run-on sentences or repeated words would help. For instance, ensure that phrases like “presence of multiple diseases” or “presence of multiple diagnosed diseases” are consistently used.

Response: The authors thank the Reviewer for the helpful suggestion. We have carefully reviewed the manuscript for consistency and clarity, ensuring that terminology is used uniformly throughout. We appreciate the Reviewer’s feedback in helping to improve the readability of our work.

5- You have mostly used “males/females,” but in a few places, “gender” creeps in. Decide on one term for consistency  for example, “sex” if referring to biological classification.

Response: The authors thank the Reviewer for this observation. We have carefully reviewed the manuscript and confirmed that 'males' and 'females' have been consistently used throughout, without instances of 'gender' in this context. However, we appreciate the suggestion and remain attentive to maintaining consistency in terminology.